# Morphological diversity in tenrecs (Afrosoricida, Tenrecidae): comparing tenrec skull diversity to their closest relatives

Sive Finlay and Natalie Cooper

School of Natural Sciences, Trinity College Dublin, Dublin, Ireland
Trinity Centre for Biodiversity Research, Trinity College Dublin, Dublin, Ireland

## ABSTRACT

It is important to quantify patterns of morphological diversity to enhance our understanding of variation in ecological and evolutionary traits. Here, we present a quantitative analysis of morphological diversity in a family of small mammals, the tenrecs (Afrosoricida, Tenrecidae). Tenrecs are often cited as an example of an exceptionally morphologically diverse group. However, this assumption has not been tested quantitatively. We use geometric morphometric analyses of skull shape to test whether tenrecs are more morphologically diverse than their closest relatives, the golden moles (Afrosoricida, Chrysochloridae). Tenrecs occupy a wider range of ecological niches than golden moles so we predict that they will be more morphologically diverse. Contrary to our expectations, we find that tenrec skulls are only more morphologically diverse than golden moles when measured in lateral view. Furthermore, similarities among the species-rich *Microgale* tenrec genus appear to mask higher morphological diversity in the rest of the family. These results reveal new insights into the morphological diversity of tenrecs and highlight the importance of using quantitative methods to test qualitative assumptions about patterns of morphological diversity.

## INTRODUCTION

Analysing patterns of morphological diversity (the variation in physical form *Foote, 1997*) has important implications for our understanding of ecological and evolutionary traits. Increasingly, many studies recognise the importance of quantifying the degree of morphological diversity instead of relying on subjective assessments of diversity in form (e.g., *Ruta et al., 2013*; *Hopkins, 2013*; *Goswami, Milne & Wroe, 2011*; *Drake & Klingenberg, 2010*; *Price et al., 2010*; *Brusatte et al., 2008*). We need to quantify the morphological similarities and differences among species to gain a better understanding of their ecological interactions and evolutionary history.

Unfortunately, morphological diversity is difficult to quantify. Many studies are constrained to measuring the diversity of specific traits rather than overall morphologies (*Roy & Foote, 1997*). In addition, our perception of morphological diversity is influenced

Corresponding author
Natalie Cooper, ncooper@tcd.ie

by the trait being measured, and results may depend on the particular trait being analysed (*Foth, Brusatte & Butler, 2012*). Furthermore, linear measurements of morphological traits can restrict our understanding of overall morphological variation; a distance matrix of measurements among specific points is unlikely to give a complete representation of a three dimensional structure (*Rohlf & Marcus, 1993*). Geometric morphometric approaches can circumvent some of these issues by using a system of Cartesian landmark coordinates to define anatomical points (*Adams, Rohlf & Slice, 2004*, and references therein). This method captures more of the true, overall anatomical shape of specific structures (*Mitteroecker & Gunz, 2009*). In particular, two-dimensional geometric morphometric approaches are commonly used to analyse 3D morphological shapes and are appropriate for cross-species comparisons (e.g., *Muschick, Indermaur & Salzburger, 2012*; *Panchetti et al., 2008*; *Wroe & Milne, 2007*). Any bias from 2D representation of a 3D structure is unlikely to be a significant issue for interspecific studies, as the overall shape variation among species is geater than discrepancies introduced by using 2D morphometric techniques (*Cardini, 2014*). These more detailed approaches are useful tools for studying patterns of morphological diversity.

Here we apply geometric morphometric techniques to quantify morphological diversity in a family of small mammals, the tenrecs. Tenrecs (Afrosoricida, Tenrecidae) are a morphologically diverse group that researchers often identify as an example of both convergent evolution and an adaptive radiation (*Soarimalala & Goodman, 2011*; *Eisenberg & Gould, 1969*). The family is comprised of 34 species, 31 of which are endemic to Madagascar (*Olson, 2013*). Body masses of tenrecs span three orders of magnitude (2.5 to >2,000 g): a greater range than all other families, and most orders, of living mammals (*Olson & Goodman, 2003*). Within this vast size range there are tenrecs which resemble shrews (*Microgale* tenrecs), moles (*Oryzorictes* tenrecs) and hedgehogs (*Echinops* and *Setifer* tenrecs, *Eisenberg & Gould, 1969*). The similarities among tenrecs and other small mammal species include examples of morphological, behavioural and ecological convergence (*Soarimalala & Goodman, 2011*). Tenrecs are one of only four endemic mammalian clades in Madagascar and the small mammal species they resemble are absent from the island (*Garbutt, 1999*). Therefore, it appears that tenrecs represent an adaptive radiation of species which filled otherwise vacant ecological niches through gradual morphological specialisations (*Poux et al., 2008*).

The claims that tenrecs are an example of both an adaptive radiation and convergent evolution have not been investigated quantitatively. There are qualitative similarities among the hind limb morphologies of tenrecs and several other unrelated species with similar locomotory styles (*Salton & Sargis, 2009*) but the degree of morphological similarity has not been established. Morphological diversity is an important feature of adaptive radiations (*Losos & Mahler, 2010*) and it also informs our understanding of convergent phenotypes (*Muschick, Indermaur & Salzburger, 2012*). Therefore, it is important to quantify patterns of morphological diversity in tenrecs to gain an insight into their evolution.

We present the first quantitative study of patterns of morphological diversity in tenrecs. We use geometric morphometric techniques (*Rohlf & Marcus, 1993*) to compare cranial morphological diversity in tenrecs to that of their closest relatives, the golden moles (Afrosoricida, Chrysochloridae). We expect tenrecs to be more morphologically diverse than golden moles because tenrecs occupy a wider variety of ecological niches. The tenrec family includes terrestrial, semi-fossorial, semi-aquatic and semi-arboreal species (*Soarimalala & Goodman, 2011*). In contrast, all golden moles occupy very similar, fossorial ecological niches (*Bronner, 1995*). Greater ecological variety is often (though not always: *McGee & Wainwright, 2013*; *Losos & Mahler, 2010*) correlated with higher morphological diversity. However, our results reveal that, in skulls at least, morphological diversity in tenrecs is not as great as it first appears.

## MATERIALS AND METHODS

Our methods involved (i) data collection, (ii) geometric morphometric analyses and (iii) estimating morphological diversity. For clarity, Fig. 1 summarises all of these steps and we describe them in detail below.

### Data collection

One of us (SF) collected data from five museums: the Natural History Museum, London (BMNH); the Smithsonian Institute Natural History Museum, Washington D.C. (SI); the American Museum of Natural History, New York (AMNH); the Museum of Comparative Zoology, Cambridge M.A. (MCZ); and the Field Museum of Natural History, Chicago (FMNH). We used the taxonomy in Wilson & Reeder's Mammal Species of the World  (*2005*), except for the recently discovered tenrec species *Microgale jobihely* (*Goodman et al., 2006*). We photographed all of the undamaged tenrec and golden mole skulls available in the collections. This included 31 of the 34 species in the tenrec family (*Olson, 2013*) and 12 of the 21 species of golden moles (*Wilson & Reeder, 2005*).

We took pictures of the skulls using photographic copy stands. To take possible light variability into account, we took a photograph of a white sheet of paper each day and used the custom white balance function on the camera to set the image as the baseline "white" measurement for those particular light conditions. We photographed the specimens with a Canon EOS 650D camera fitted with a EF 100 mm f/2.8 Macro USM lens and using a remote control (Hähnel Combi TF; Hahnel, Cork, Ireland) to avoid camera shake. We photographed the specimens on a black material background with a light source in the top left-hand corner of the photograph and a scale bar placed below the specimen. We used small bean bags to hold the specimens in position to ensure that they lay in a flat plane relative to the camera, and used the grid-line function on the live-view display screen of the camera to position the specimens in the centre of each image.

All skulls were photographed in three views: dorsal, ventral and lateral (right side) (Fig. 1). When the right sides of the skulls were damaged or incomplete, we photographed the left sides and later reflected the images (e.g., *Barrow & Macleod, 2008*). Some specimens were too damaged to use in particular views so there were a different total number of images for each analysis. Our final data sets included photographs of 182 skulls in dorsal
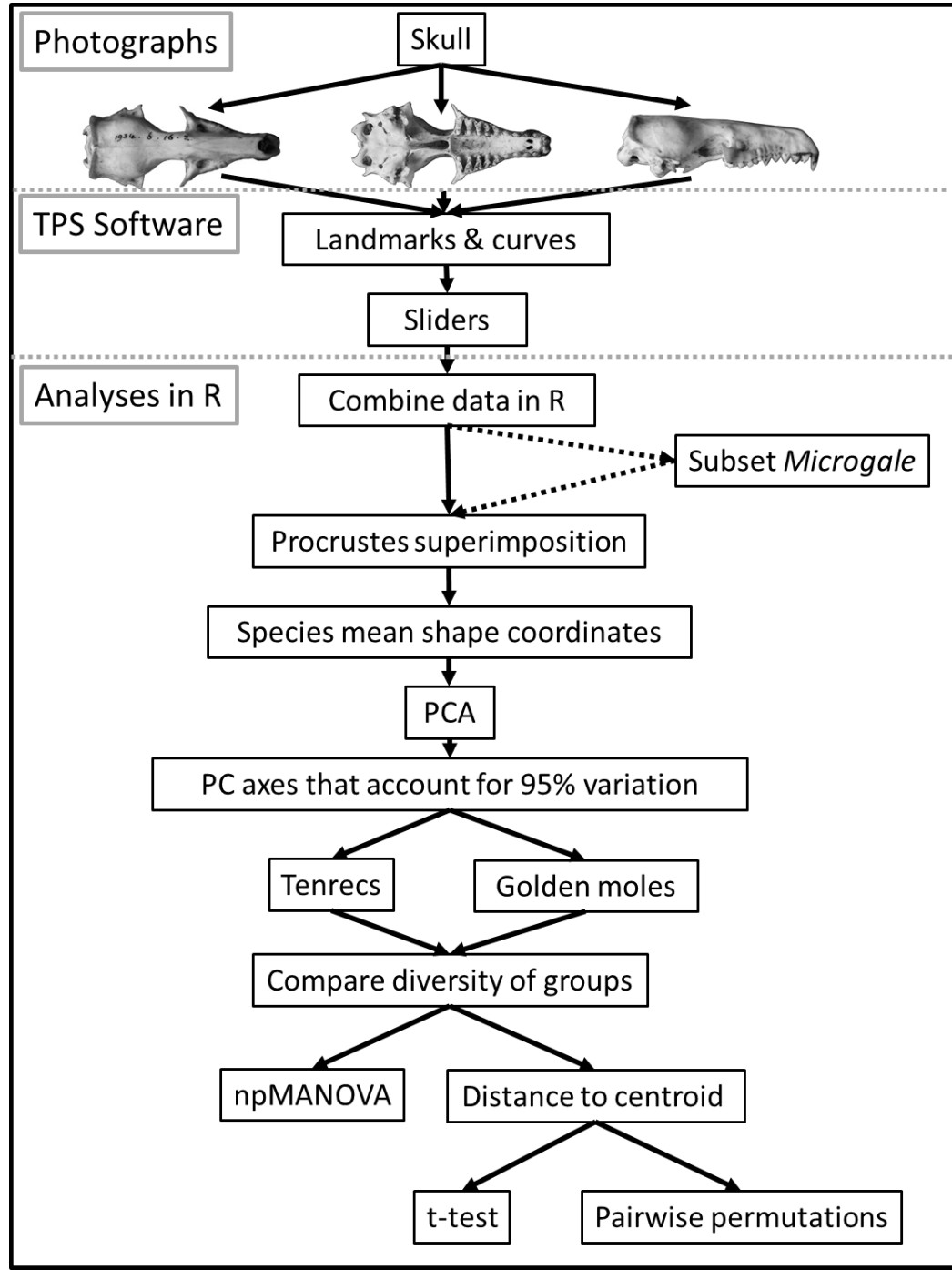

**Figure 1 Flowchart diagram of data collection and analysis.** Summary of the main steps in our data collection, processing and analysis protocol. Note that the analyses were repeated separately for each set of photographs: skulls in dorsal, ventral and lateral views. The dashed arrows refer to the stage at which we selected a subsample of the tenrecs (including just five species of the *Microgale* genus) so that we could compare the morphological diversity of this reduced subsample of tenrec species to the diversity of golden moles.

view (148 tenrecs and 34 golden moles), 173 skulls in ventral view (141 tenrecs and 32 golden moles) and 171 skulls in lateral view (140 tenrecs and 31 golden moles). Details of the total sample size for each species can be found in Supplemental Information.

After taking the photographs we used the Canon Digital Photo Professional software (*Canon, 2013*) to convert the raw files to binary (grey scale) images and re-save them as TIFF files (uncompressed files preserve greater detail, *RHOI, 2013*). Photographs of the specimens from the American Museum of Natural History and the Smithsonian Institute Natural History Museum are available on figshare (dorsal; *Finlay & Cooper (2013a)*, ventral; *Finlay & Cooper (2013c)* and lateral; *Finlay & Cooper (2013b)*). Copyright restrictions from the other museums prevent public sharing of their images but they are available from the authors on request.

## Geometric morphometric analyses

We used a combination of landmark and semilandmark approaches to assess the shape variability in the skulls. We used the TPS software suite (*Rohlf, 2013*) to digitise landmarks and curves on the photos. We set the scale on each image individually to standardise for the different camera heights used when photographing the specimens. We created separate data files for each of the three morphometric analyses (dorsal, ventral and lateral views). One of us (SF) digitised landmarks and semilandmark points on every image individually.

Figure 2 depicts the landmarks and curves which we used for each skull view. For landmarks defined by dental structures, we used published dental sources where available (*Repenning, 1967*; *Eisenberg & Gould, 1969*; *Nowak, 1983*; *MacPhee, 1987*; *Knox Jones & Manning, 1992*; *Davis & Schmidly, 1997*; *Quérouil et al., 2001*; *Nagorsen, 2002*; *Wilson & Reeder, 2005*; *Goodman et al., 2006*; *Karataş, Mouradi Gharkheloo & Kankiliç, 2007*; *Hoffmann & Lunde, 2008*; *Asher & Lehmann, 2008*; *Muldoon et al., 2009*; *Lin & Motokawa, 2010*) to identify the number and type of teeth in each species. Detailed descriptions of the landmarks can be found in the Supplemental Information (Tables S1, S2, S3) along with an example figure of landmarks on golden mole skulls (Fig. S1).

When using semilandmark approaches there is a potential problem of over-sampling: simpler structures will require fewer semilandmarks to accurately represent their shape (*MacLeod, 2012*). To ensure that we applied a uniform standard of shape representation to each outline segment (i.e., that simple structures would not be over-represented and more complex features would not be under-represented), we followed the method outlined by *MacLeod (2012)*. This re-sampling method determines the minimum number of semilandmark points required to measure an outline length to at least 95% accuracy of the true length of the outline. The procedure balances the need to represent outline shapes accurately without introducing error by over-sampling curves (*MacLeod, 2012*). We used 54 points for skulls in dorsal view (10 landmarks, 44 semilandmarks across 4 curves), 73 points for skulls in ventral view (13 landmarks, 60 semilandmarks) and 44 points for skulls in lateral view (9 landmarks and 35 semilandmarks across 2 curves). See Fig. 2 and Supplemental Information for more details.

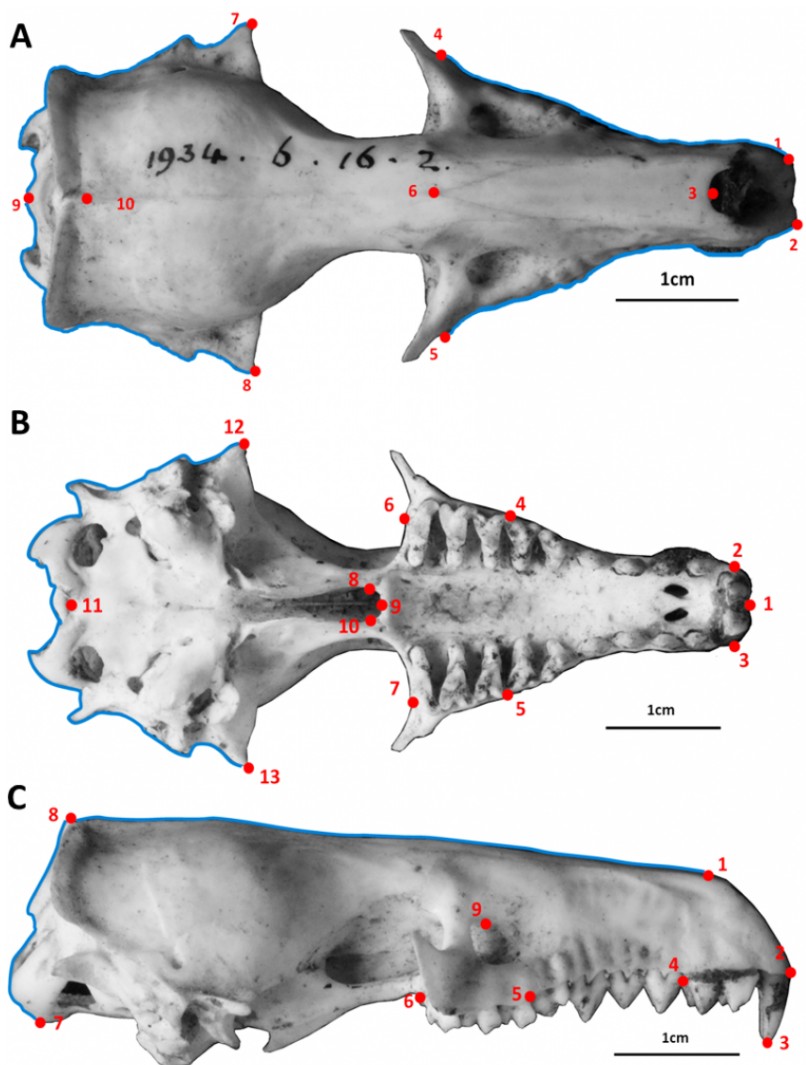

**Figure 2 Skulls: dorsal, ventral and lateral landmarks.** Landmarks (numbered points) and curves (outlines) for the skulls in dorsal (A),ventral (B) and lateral (C) view. See the Supplemental Information for detailed landmark descriptions. The skulls are an example of a *Potamogale velox* (otter shrew tenrec), museum accession number BMNH 1934.6.16.2.

After creating the files with the landmarks and semilandmarks placed on each photograph, we used TPSUtil (*Rohlf, 2012*) to create "sliders" files that defined which points in the TPS files should be treated as semilandmarks (*Zelditch, Swiderski & Sheets, 2012*). We combined the landmarks and taxonomic identification files into a single morphometrics data object and carried out all further analyses in R version 3.1.1 (*R Core Team, 2014*).

Next we used the gpagen function in version 2.1 of the geomorph package (*Adams et al., 2014*; *Adams & Otárola-Castillo, 2013*) to run a general Procrustes alignment (*Rohlf & Marcus, 1993*) of the landmark coordinates while sliding the semilandmarks by minimising Procrustes distance (*Bookstein, 1997*). We used these Procrustes-aligned coordinates of all specimens to calculate average shape values for each species which we then used for

a principal components (PC) analysis with the `plotTangentSpace` function (*Adams & Otárola-Castillo, 2013*). We selected the number of principal component (PC) axes that accounted for 95% of the variation in the data (Fig. 1) and used these axes to estimate morphological diversity in each family.

The majority of tenrec species (19 out of 31 in our data) belong to the *Microgale* (shrew-like) genus that has relatively low morphological diversity (*Soarimalala & Goodman, 2011*; *Jenkins, 2003*). This may mask signals of higher morphological diversity among other tenrecs. To test this, we created a subset of the tenrec data that included just five of the *Microgale* species, each representing one of the five sub-divisions of *Microgale* outlined by *Soarimalala & Goodman (2011)*, i.e., small, small-medium, medium, large and long-tailed species. We repeated the general Procrustes alignment described above using this reduced data set. We then compared the morphological diversity of the full data set (31 species of tenrec) or the reduced data set with just 17 species of tenrec (five *Microgale* and 12 non-*Microgale* species; Fig. 1) to that of the 12 species of golden moles.

## Estimating morphological diversity

We grouped the PC scores for tenrecs and golden moles separately so that we could estimate the diversity of each family and then compare the two groups (Fig. 1). We compared morphological diversity in two ways. First, we used non parametric multivariate analysis of variance (npMANOVA; *Anderson, 2001*) to test whether tenrecs and golden moles occupied significantly different positions within the morphospaces defined by the PC axes that accounted for 95% of the overall variation in the data (e.g., *Serb et al., 2011*; *Ruta et al., 2013*). A significant difference between the two families would indicate that they have unique morphologies which do not overlap. Second, we compared morphological diversity within tenrecs to the diversity within golden moles.

Morphological diversity (variation in form) is more commonly referred to as morphological disparity (*Foote, 1997*). There are many different methods for measuring disparity. Calculations based on summary (principal component) axes of shape variation are popular (e.g., *Ruta et al., 2013*; *Foth, Brusatte & Butler, 2012*; *Brusatte et al., 2008*; *Wainwright, 2007*) while other methods include calculating disparity directly from Procrustes shape variables (*Zelditch, Swiderski & Sheets, 2012*) or rate-based approaches which depend on phylogenetic branching patterns (e.g., *Price et al., 2013*; *Price et al., 2010*; *O'Meara et al., 2006*). There is no single best method of measuring disparity (*Ciampaglio, Kemp & McShea, 2001*) and each method makes different assumptions which are appropriate for different situations. Therefore, for clarity, we have chosen to measure variation in physical form using a clear, easily-interpretable method which captures variation in morphological diversity.

We define morphological diversity as the mean Euclidean distance (sum of squared differences) between each species and its family centroid (Fig. 3). This is summarised in the equation below where $n$ is the number of species in the family, $i$ is the number of PC axes
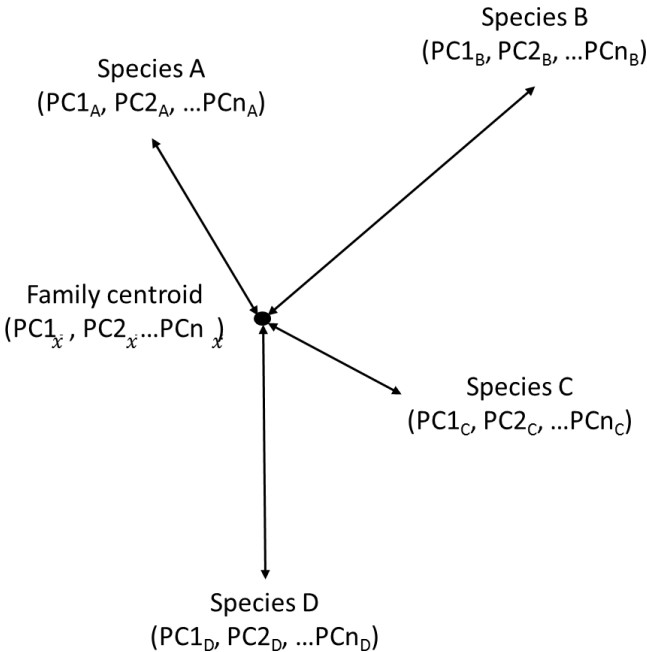

**Figure 3** **Calculating diversity as mean Euclidean distance to Family centroid.** Estimating morphological diversity as the mean Euclidean distance between each species and the family centroid. Every species had scores on the principal components (PC) axes that accounted for 95% of the variation in the principal components analysis. The number of axes (PC$n$) varied for each analysis but they were the same within a single analysis. PC scores were used to calculate the Euclidean distance from each species to the family centroid (average PC scores for the entire family). Morphological diversity of the family is the average value of these Euclidean distances.

and $c$ is the average PC score for each axis (the centroid).

$$\text{Diversity} = \frac{\sqrt{\Sigma(\text{PC}n_i - \text{PC}c_i)^2}}{n}. \tag{1}$$

If tenrecs are more morphologically diverse than golden moles, then they should be more dispersed within the morphospaces and have, on average, higher values of mean Euclidean distance.

One possible issue with these analyses is that the two families have unequal sample sizes: 31 (or a subset of 17) tenrec species compared to just 12 golden mole species. Morphological diversity is usually decoupled from taxonomic diversity (e.g., *Ruta et al., 2013*; *Hopkins, 2013*) so larger groups are not necessarily more morphologically diverse. However, comparing morphological diversity in tenrecs to the diversity of a smaller family could still bias the results. We used pairwise permutation tests to account for this potential issue.

We tested the null hypothesis that tenrecs and golden moles have the same morphological diversity (the same mean Euclidean distance to the family centroid). If this is supported, when we randomly assign the group identity of each species (i.e., shuffle the "tenrec" and "golden mole" labels) and then re-compare the morphological diversity of the two groups, there should be no significant difference between these results and those obtained when the species are assigned to the correct groupings.

We performed this shuffling procedure (random assignation of group identity) 1,000 times and calculated the difference in morphological diversity between the two groups for each permutation. This generated a distribution of 1,000 values which are calculations of the differences in morphological diversity under the assumption that the null hypothesis (equal morphological diversity in the two families) is true. This method automatically accounts for differences in sample size because shuffling of the group labels preserves the sample size of each group: there will always be 12 species labelled as "golden mole" and then, depending on the analysis, either 31 or 17 species labelled as "tenrec." Therefore, the 1,000 permuted values of differences in morphological diversity create a distribution of the expected difference in diversity between a group of sample size $N = 31$ (or $N = 17$ in the case of the tenrec data subset) compared to a group of sample size $N = 12$ under the null hypothesis that the two groups have the same morphological diversity. We compared the observed measures of the differences in morphological diversity between the two families to these null distributions to determine whether there were significant differences after taking sample size into account (two-tailed $t$ test). Data and code for all of our analyses are available on GitHub (*Finlay & Cooper, 2015*).

## RESULTS

Figure 4 depicts the morphospaces defined by the first two principal component (PC) axes from our principal components analyses (PCAs) of skull and mandible morphologies. The PCAs are based on the average Procrustes-superimposed shape coordinates for skulls in three views (dorsal, ventral and lateral).

To compare morphological diversity in the two families, we used the PC axes which accounted for 95% of the cumulative variation in each of the skull analyses: dorsal ($n = 6$ axes), ventral ($n = 7$ axes) and lateral ($n = 7$ axes). First, we compared the position of each family within the morphospace plots. Tenrecs and golden moles occupy significantly different positions in the dorsal (npMANOVA: $F_{1,42} = 68.13$, $R^2 = 0.62$, $p = 0.001$), ventral (npMANOVA: $F_{1,42} = 103.33$, $R^2 = 0.72$, $p = 0.001$) and lateral (npMANOVA: $F_{1,42} = 76.7$, $R^2 = 0.65$, $p = 0.001$) skull morphospaces, indicating that the families have very different, non-overlapping cranial and mandible morphologies (Fig. 4). For each analysis, PC1 summarises a morphological change from the foreshortened, "squat" shape of golden mole skulls at one extreme to the rostrally elongated shape of tenrecs (particularly the *Microgale*) at the other extreme.

Second, we compared the morphological diversity within each family. Based on our measures of mean Euclidean distance to the family centroids, tenrec skulls are more morphologically diverse than golden mole skulls when they are measured in lateral view but not in dorsal or ventral view (Table 1). In contrast, when we analysed morphological diversity of skulls within the sub-sample of 17 tenrecs (including just five *Microgale* species) compared to the 12 golden mole species, we found that tenrec skulls were significantly more morphologically diverse than golden moles in all analyses (Table 1). The pairwise permutation tests for each analysis confirmed that differences in morphological diversity were not artefacts of differences in sample size (Table 2).

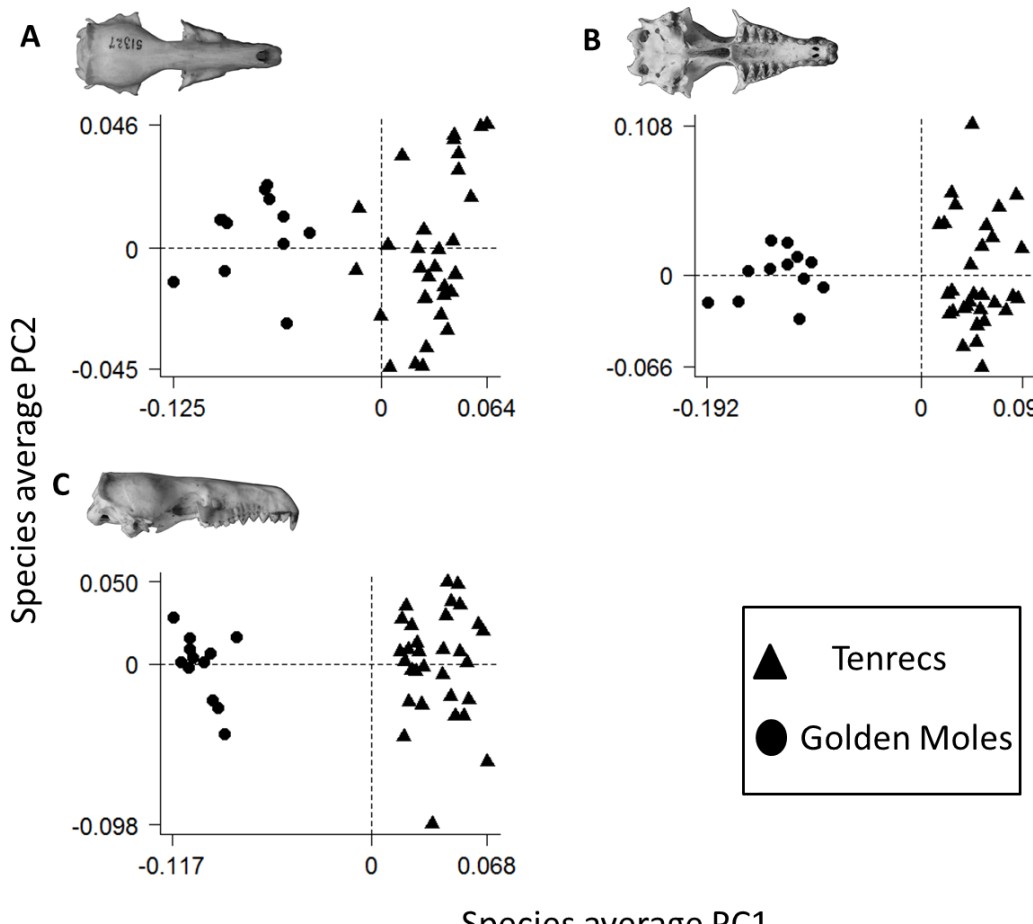

**Figure 4 Morphospace (principal components) plot of morphological diversity in tenrec and golden mole skulls.** Principal components plots of the morphospaces occupied by tenrecs (triangles, $n = 31$ species) and golden moles (circles, $n = 12$ species) for skulls in dorsal (A), ventral (B) and lateral (C) views. Each point represents the average skull shape of an individual species. Axes are principal component 1 (PC1) and principal component 2 (PC2) of the average scores from principal components analyses of mean Procrustes shape coordinates for each species.

## DISCUSSION

Tenrecs are often cited as an example of a mammalian group with high morphological diversity (*Olson, 2013*; *Soarimalala & Goodman, 2011*; *Eisenberg & Gould, 1969*). They are also more ecologically diverse than their closest relatives (*Soarimalala & Goodman, 2011*; *Bronner, 1995*) so we predicted that they would be more morphologically diverse than golden moles. However, our results do not support our original prediction, highlighting the importance of quantitative tests of perceived morphological patterns.

In our full analysis, tenrecs only had higher morphological diversity than golden moles when the skulls were measured in lateral view (Table 1). There was no difference in morphological diversity when we analysed the skulls in dorsal or ventral views. This is most likely due to our choice of landmarks. The two outline curves in lateral view (Fig. 2) emphasise morphological variation in the back and top of the skulls. These curves

**Table 1 Comparing morphological diversity in tenrecs and golden moles.** Morphological diversity in tenrecs compared to golden moles (12 species). $N$ is the number of tenrec species: 31 species or 17 species including just five representatives of the *Microgale* genus. Morphological diversity of the family is the mean Euclidean distance from each species to the family centroid. Significant differences between the two families ($p < 0.05$) from two-tailed $t$-tests are highlighted in bold.

| $N$ | Analysis | Morphological diversity | | $t_{df}$ | $p$ value |
|---|---|---|---|---|---|
| | | Tenrecs (mean ± s.e) | Golden moles (mean ± s.e) | | |
| 31 | Skulls dorsal | 0.036 ± 0.0029 | 0.029 ± 0.0032 | $-1.63_{29.88}$ | 0.11 |
| | Skulls ventral | 0.048 ± 0.0034 | 0.044 ± 0.0041 | $-0.68_{26.99}$ | 0.51 |
| | Skulls lateral | 0.044 ± 0.0041 | 0.032 ± 0.0037 | $-2.16_{35.03}$ | **0.04** |
| 17 | Skulls dorsal | 0.044 ± 0.0025 | 0.029 ± 0.0032 | $-3.62_{22.75}$ | **<0.01** |
| | Skulls ventral | 0.054 ± 0.0039 | 0.042 ± 0.0041 | $-2.23_{25.46}$ | **0.04** |
| | Skulls lateral | 0.054 ± 0.0053 | 0.031 ± 0.0037 | $-3.47_{26.31}$ | **<0.01** |

**Table 2 Results of the permutation tests.** Results of the permutation analyses comparing the observed differences in morphological diversity to a null distribution of expected results. Morphological diversity of the family is the mean Euclidean distance from each species to the family centroid. Results are shown for both the full ($N = 31$ species of tenrec compared to 12 species of golden mole) and reduced ($N = 17$ species of tenrec compared to 12 golden moles) data sets. Significant values ($p < 0.05$) indicate that the observed morphological diversity is different to the expected differences under a null hypothesis of equivalent diversities in the two families.

| $N$ | Analysis | Morphological diversity | | | | | $p$ value |
|---|---|---|---|---|---|---|---|
| | | Measured values | | | Permuted values | | |
| | | Tenrecs | Golden moles | Difference | Min. | Max. | |
| 31 | Dorsal | 0.036 | 0.029 | 0.007 | −0.011 | 0.009 | **0.013** |
| | Ventral | 0.048 | 0.044 | 0.004 | −0.014 | 0.013 | **0.023** |
| | Lateral | 0.044 | 0.032 | 0.012 | −0.012 | 0.011 | **<0.001** |
| 17 | Dorsal | 0.044 | 0.029 | 0.015 | −0.011 | 0.014 | **<0.001** |
| | Ventral | 0.054 | 0.042 | 0.013 | −0.017 | 0.019 | **0.023** |
| | Lateral | 0.054 | 0.031 | 0.022 | −0.018 | 0.019 | **<0.001** |

summarise overall shape variation but they do not identify clear anatomical differences because they are defined by relative features rather than homologous structures (*Zelditch, Swiderski & Sheets, 2012*). Therefore, high morphological diversity in tenrecs when analysed in this view may not indicate biologically or ecologically relevant variation. These lateral aspects of the skull morphology were not visible in the dorsal and ventral photographs so they could not be included in those analyses. In contrast, our landmarks in the dorsal, and particularly ventral, views focus on morphological variation in the overall outline shape of the sides of the skull and palate (Fig. 2). The result that tenrecs are no more diverse than golden moles in these areas makes intuitive sense: most tenrecs have non-specialised insectivorous or faunivorous diets (*Olson, 2013*) so there is no obvious functional reason why they should have particularly diverse palate morphologies.

Similarly, while there are anatomical differences among tenrec tooth morphologies (*Asher & Sánchez-Villagra, 2005*) more work is required to determine if and how those differences correspond to variation in diet or feeding ecology. The different results for our analysis of lateral skull morphologies compared to dorsal and ventral views highlight the importance of using multiple approaches when studying 3D morphological shape using 2D geometric morphometrics techniques (*Arnqvist & Mårtensson, 1998*). Future analyses could use 3D geometric morphometric approaches to test whether similar patterns emerge.

Landmark choice and placement will inevitably influence the results of a geometric morphometrics study. Our interest in broad-scale, cross-taxonomic comparisons of cranial morphology constrained our choice of landmarks to those that could be accurately identified in many different species (e.g., *Ruta et al., 2013*; *Goswami, Milne & Wroe, 2011*; *Wroe & Milne, 2007*; *Goswami, 2006*). In contrast, studies that use skulls to characterise morphological variation within species (e.g., *Blagojević & Milošević-Zlatanović, 2011*; *Giannini et al., 2010*; *Flores, Abdala & Giannini, 2010*; *Bornholdt, Oliveira & Fabián, 2008*) or to delineate species boundaries within a clade (e.g., *Panchetti et al., 2008*) tend to focus on more detailed, biologically homologous landmarks (*Zelditch, Swiderski & Sheets, 2012*). Repeating our analyses with a narrower taxonomic focus may give greater insight into the specific morphological differences among subgroups of tenrecs and golden moles.

In addition to the differences among the three skull views, our results indicate that, in skulls at least, the overall morphological diversity within tenrecs is not as large as is often assumed (e.g., *Eisenberg & Gould, 1969*; *Olson, 2013*). Studies of morphological variation are sensitive to the sampling used. If a particular morphotype is over-represented then the similarities among those species will reduce the overall morphological variation within the group (*Foote, 1991*). This appears to be the case for our data; it was only when we included a sub-sample of *Microgale* tenrecs that we found higher morphological diversity in tenrecs compared to golden moles across all three skull analyses (Table 1). While there are clear physical differences among family members (*Olson, 2013*; *Eisenberg & Gould, 1969*), the majority of tenrecs (the *Microgale*) are very morphologically similar (*Jenkins, 2003*) so morphological diversity in the family as a whole is not as large as it first appears.

The goal of our study was to quantify morphological variation in tenrecs instead of relying on subjective assessments of their high morphological diversity. However, it is difficult to quantify overall morphological diversity because any study is inevitably constrained by its choice of specific traits (*Roy & Foote, 1997*). While the skull is widely regarded as a good model for studying morphological variation (e.g., *Blagojević & Milošević-Zlatanović, 2011*; *Flores, Abdala & Giannini, 2010*; *Giannini et al., 2010*), quantifying variation in other morphological traits could yield different patterns. Therefore future work should extend our approach beyond skulls to gain a more complete understanding of the overall morphological diversity of tenrecs and golden moles. While recognising these limitations, our results provide valuable insights into the differences between subjective and quantitative assessments of morphological diversity.

## CONCLUSIONS

We have presented the first quantitative investigation of morphological diversity in tenrecs. Our results indicate that, overall, tenrec skulls are not more morphologically diverse than golden moles and that similarities among the species rich *Microgale* tenrecs mask signals of higher morphological diversity among the rest of the family. Of course, the results presented here are restricted to just one aspect of morphological variation and further analysis of other traits is required. However, our findings provide a foundation for future investigations and represent a significant step towards a more quantitative understanding of patterns of morphological and evolutionary diversity in tenrecs.

## ACKNOWLEDGEMENTS

We thank Thomas Guillerme, François Gould and the members of NERD club for their insightful discussions and comments. We are grateful to the Editor, Nick Crumpton and an anonymous reviewer for their constructive comments. Thank you to museum staff and curators for facilitating our access to their collections: Leona Leonard and Nigel Monaghan (Natural History Museum, Ireland), Roberto Portela Miguez and Paula Jenkins (Natural History Museum, London), Esther Langan (Smithsonian Institute NMNH), Eileen Westwig (American Museum of Natural History), Judy Chupasko (Museum of Comparative Zoology) and Bill Stanley and Steve Goodman (Field Museum).

### Funding

This work was funded by an Irish Research Council EMBARK Initiative Postgraduate Scholarship (SF) and a European Commission CORDIS Seventh Framework Programme (FP7) Marie Curie CIG grant (proposal number: 321696; NC). The funders had no role in study design, data collection and analysis, decision to publish, or preparation of the manuscript.

### Grant Disclosures

The following grant information was disclosed by the authors:
Irish Research Council EMBARK Initiative Postgraduate Scholarship.
European Commission CORDIS Seventh Framework Programme.
Marie Curie CIG: 321696.

### Competing Interests

The authors declare there are no competing interests.

### Author Contributions

- Sive Finlay conceived and designed the experiments, performed the experiments, analyzed the data, wrote the paper, prepared figures and/or tables, reviewed drafts of the paper.
- Natalie Cooper conceived and designed the experiments, wrote the paper, reviewed drafts of the paper.

**Peer**J

## Data Deposition

The following information was supplied regarding the deposition of related data:

(1) "Insectivore" mammal skulls, dorsal view.
http://dx.doi.org/10.6084/m9.figshare.705863.

(2) "Insectivore" mammal skulls, lateral view.
http://dx.doi.org/10.6084/m9.figshare.715890.

(3) "Insectivore" mammal skulls, ventral view.
http://dx.doi.org/10.6084/m9.figshare.715841.

## Supplemental Information

Supplemental information for this article can be found online at http://dx.doi.org/10.7717/peerj.927#supplemental-information.

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
