# Peer review of "Morphological diversity in tenrecs (Afrosoricida, Tenrecidae): comparing tenrec skull diversity to their closest relatives"

_PeerJ, doi:10.7717/peerj.927_

## Round 0.1 · original submission · Minor Revisions

Overall, I think this is an excellent, well presented study that will be suitable for publication in Peer J following minor revisions. Both reviewers share a similar, positive view of the manuscript and have provided some helpful suggestions that I ask the authors to please consider carefully when revising their manuscript. Particularly,

1. Reviewer #1 has noted that the bibliography could be improved, and that a number of references are duplicated throughout the text. I agree with this statement, and see several areas where the authors could qualify their statements with stronger referencing. For example, the statement that tenrecs are commonly considered an example of adaptive radiation, needs to be better qualified, or wording needs to be amended accordingly.
2. Reviewer #1 has also made some important suggestions regarding improving aesthetic of the Figures. Similarly, Reviewer #2 has suggested Fig. 1 could be moved to supplementary information given the extensive details provided on the method (though currently sample size is ommitted?)
3. Reviewer #2 has made an important point regarding the use of “morphological diversity” rather than “disparity” and suggested some key citations that should be included. I am in strong agreement with their statement that at present the authors have over stated the lack of quantitative approach in this area – please check those comments and revisit that statement. Doing so will not detract from the importance of this article.
4. Reviewer #2 has queried the approach used to quantify disparity, I agree with their comment here, and ask the authors to please either consider extending their analytical approach or providing further justification for their choice of method

In addition to the reviewer comments, please find below some additional points from my review of the manuscript. Importantly, I suggest that the authors should please reconsider to revise their first paragraph, which at the moment does not lead into the paper, or do justice to the importance of your study.

Specific points:

Abstract
I think that the first statement is not accurate. There exist a large number of studies on morphological diversity (disparity) that encapsulate both extant and extinct groups.
Introduction
Pg2, ln8-18. I think this first paragraph is generally confused in that the authors deal with research areas that are vast within very few qualified sentences that do not relate directly to the topic in hand. For example, adaptive radiation is not something studied here, (or if it is, then it needs to be signaled much earlier in the paper, directly and with numerous examples for tenrecs). The references do not include classical works such as Schluter 2000, and could easily comprise many review papers (e.g. Gavrilets and Losos 2009;Salzburger 2009; Salzburger et al. 2014; Kocher 2004; Gavrilets and Vose 2005…) rather than a repetition of Olson and Arroyo-Santos 2009.
In a related matter, morphological convergence is introduced without a clear explanation about its relationship to adaptive radiation, or ecomorphological variation, niche exploitation.
I suggest that the authors re-phrase this paragraph to better introduce their study. Also, somewhere you should provide a definition of morphological diversity (variation in form) or more commonly referred as, disparity. Consider citing Foote.
Pg2, ln22: I think this is a slight over statement, there are really many studies of morphological diversity (disparity). Please see for example papers by Drake and Klingenberg (dogs), Gerber, Polly, Weisbecker, Sears…
Pg2, ln29: OK, but see e.g. Polly and MacLeod eigensurface, or papers on statistical atlases (e.g. Fatah et al. 2012 AJPA) that allow entire shapes to be appreciated rather than single traits.
Pg3, ln46: the references cited here do not reflect the statement – I would expect to find cichlids, anoles, icefishes, stickelbacks, Darwin’s finches in any recent review or book on adaptive radiation, could you cite similar references for tenrecs? Also, see Poux et al. 2008 – BMC Evol Biol and their statements on diversification rates in tenrecs. - doi:10.1186/1471-2148-8-102
Pg6, ln101: please could you provide specimen numbers here – you mention that much later (ln140), but I think it would be helpful for the reader to have that information right away.
Pg7, ln141: should be “the” rather than “that”
Figure 2: I suggest the authors use colour here. For instance, you could colour the landmarks e.g. red, which would help distinguish those against the photographs. At present it is difficult to see the landmarks easily.
Pg14, ln304: agreed, however it might also equally suggest that a complete 3D GMM approach would also be fitting.
Pg15, ln310 – you may also want to check Goswami’s early papers (e.g. 2006 in Am Nat; 2007 in PLoS ONE) in which she defines a subset of homologous landmarks for a wide variety of mammalian clades.
Pg312: I think you could add citations for papers by Norberto Giannini or David Flores here (extensive work on cranial anatomy)
Pg16, ln3338: agreed, but I think you put your study down here! The skull is an excellent model that shows a high diversity in form related to function, and that has been widely studied.
Pg16, ln345: another suggestion might be to consider exploring the ontogenetic basis for differing levels of morphological diversity? For example, check papers by Daisuke Koyabu on mammalian cranial development. (Nat Comm, PNAS)

Reviewer 1 ·

Basic reporting

The manuscript is well written and well presented. The first two sentences of the abstract, however, do not do the manuscript justice. The first sentence of the introduction has a better hook, so I suggest using something like that.

The only issue I have in the basic reporting is regarding the comment that nowadays morphological diversity is predominantly studied qualitatively – the palaeobiology and evolutionary literature is rife with studies of quantitative disparity (morphological diversity). Thus I disagree that there are “few examples” (line 21). The more interesting foci for why this study was done would be to talk about why morphological diversity differs among clades, what it is and isn’t related to and what it means to study it.

Experimental design

The study is clearly outlined and well designed. The sampling efforts are seemingly impressive, covering 43 species out of 55 species from the two families under study, however the number of specimens per species is not fully disclosed; only a total of specimens for each view. I suggest a supplementary table summarising how many specimens per species.
The supplementary materials text describes the authors’ diligence in measurement error from digitizing and photographing 3D objects. I applaud their conscientiousness.
The geometric morphometric analyses have mostly been carried out appropriately. I am concerned by the potentially overkill number of variables in this dataset. The authors should, but do not disclose the total number in the main text, and Figure 2 only shows the fixed landmarks, not the semilandmarks used. From the supplementary materials, I calculate there were for the dorsal view 54 landmarks and semilandmarks, for the ventral view 73 landmarks and semilandmarks and lateral view 44 landmarks and semilandmarks. For 2D data the authors are aware of issues with oversampling (lines 151-153), yet I would be wary that this dataset is indeed oversampling. For the interests of leading by example, I suggest the authors examine the number of principal components that are near zero and reassess the number of semilandmarks used. In particular, it would helpful to show by mantel tests that the distribution of species in the PCA morphospace are not substantially changed by altering the number of landmarks used. Perhaps oversampling has added variation?
Fortunately, the statistical analyses of morphological disparity, as measured by the amount of shape space occupied, are not directly affected by the greater number of variables than specimens because they use distances between species (thus taking advantage of the Q-mode R mode equivalency, Gower 1966). Therefore my concerns above with oversampling are more about making sure that having so many semilandmarks is not introducing error and thus more shape variation.


There are several methods in the literature for quantifying morphological diversity/disparity. The authors chose an average centroid size approach, which is taking the square root of the summed square distances of points to the group centroid (center of gravity) and dividing by the number of points in the group. This approach measures the cumulative disparity of points and allows for comparison of groups with different sample sizes. The pros and cons of this approach are: centroid size is the square root of within-group Sums of Squares, and is, therefore, a direct measure of within-group variation. While centroid size is hard to compare among groups with different sample sizes, by taking the average Centroid size allows centroid size to be compared among groups of different size. But, it is not really an average since centroid size is the square root of the summed squared distances, dividing by the number of points only scales this single value. Thus this measure is good as a relative scale; it makes less sense on an absolute scale.
Why did the authors choose to calculate morphological diversity this way and not using the Procrustes variance or convex hull (e.g. Drake and Klingenberg 2010 Am Nat). How do you think your results would differ if one of these was used? I’m very surprised to see a paper studying disparity that does not reference more of the classic literature, e.g. Foote 1997 Annu. Rev. Ecol. Syst., Foote 1992 Paleobiology , Ciampaglio et al. 2001 Paleobiology.

Validity of the findings

The data presented are robust and statistically sound.

Additional comments

The term “morphological diversity” is used here to refer to morphological disparity which I believe is the more commonly used term. I suggest using disparity at least once, or throughout, to aid this paper being found in computer literature searching.

I would like to see described in the results the biology behind the PC axes, i.e. what shape change the first two PCs of each view describes. Particularly since PC1 delimits the two taxa.

Figure 1 is very nicely made. However I do not see that there is any reason for its inclusion, given all of this information is described in the methods. If the author has such strong attachment to it, I suggest it is put in supplementary materials.
Figure 2 is well-presented but very dark. Suggest changing the brightness and using colour of the digitized landmark positions so that it is easier to read. Also, the semilandmarks should be depicted on this figure.
Figure 3 would benefit from colour.

The citation for the R package geomorph is wrong, Emmanuel Paradis is not an author on paper. The proper citation is:
Adams, D. C., and E. Otárola-Castillo. 2013. geomorph: an r package for the collection and analysis of geometric morphometric shape data. Methods in Ecology and Evolution 4:393-399.

Also, given that software is regularly changed and updated, please cite the version of geomorph used as follows:
Adams, D. C., E. Otarola-Castillo. 2013. geomorph: Software for geometric morphometric analyses. R package version 1.0: cran.r-project.org/web/packages/geomorph/index.html.
or
Adams, D. C., E. Otarola-Castillo, and E. Sherratt. 2014. geomorph: Software for geometric morphometric analyses. R package version 2.0: cran.r-project.org/web/packages/geomorph/index.html.
or
Adams, D. C., M. L. Collyer, E. Otarola-Castillo, and E. Sherratt. 2014. geomorph: Software for geometric morphometric analyses. R package version 2.1: cran.r-project.org/web/packages/geomorph/index.html.

·

Basic reporting

This is a very interesting, novel investigation into how morphological diversity can be studied and raises important questions about certain methodological techniques.
The article is, on the whole, written very well, although there are a number of grammatical problems and a few errors that I have highlighted in general comments.
The article is a fine structure, although a slightly broader bibliography would be of use to the reader. But overall, this is an excellent addition to the literature.

Experimental design

No comment: research questions were defined well, and investigated rigorously with appropriate techniques and an impressively large data set.

Validity of the findings

No comment: findings are clearly set out in the context of the original questions and concluded in a sufficient, easily understandable way.

Additional comments

Throughout the paper, the term 'Afrosoricida' is used to denote the tenrecid-chrysochlorid clade. It is recommended the authors consider using the term 'Tenrecoidea' for this clade. It is the authors' choice, but Asher and Helgen (2010) http://link.springer.com/article/10.1186/1471-2148-10-102 may be of interest for a detailed history of the prevailing (and relic) nomenclature for tenrecs and golden moles.

Abstract:
The term ‘exceptional’ is used twice in the first and second paragraphs and feels slightly repetitive.
Final paragraph:
First line: ‘We’ should not be capitalized.
The terms ‘genus’ and ‘family’ should not be capitalized (and throughout the paper).

[Page 2, line 18] Losos (2011) is cited as exploring ‘relative’ repeatability of evolution. However, this is after a 2012 and a 2013 reference are cited. Also, the ‘repeatability of evolution’ is an odd phrase. Do the authors mean the repeated evolution of certain morphologies due to certain (e.g. developmental) constraints? Perhaps this needs a slight tweak.

[Page 2, line 23] Examples of papers ‘qualitatively’ assessing morphological diversity would be useful here, especially as this work seeks to show qualitative work can be bettered.

[Page 2, line 19 – Page 3, line 26] Five out of the six sentences in this section begin with conjunctions and similar words (Although… However… Unfortunately… In addition… Furthermore) which sets up an unnecessarily apologetic tone.

[Page 2, line 30] ‘…influenced by the trait being used’. I recommend changing the word ‘used’ to something like ‘measured’ or ‘analysed’ earlier in the sentence. The term ‘used’ led me initially to think this would begin a section on the ‘use’ of an adaptation, e.g. functional role.

[Page 3, line 34] ‘…unlikely to give a completely accurate representation’. It seems to me that the 'accuracy' could be ‘complete’ regardless the number of measurements. Even a few measurements could still be ‘accurate’ depending on their repeatability. The resolution of the 3-D shape’s description is perhaps the concept the authors are getting at? Although I’m sure there’s a better way to say that.

[Page 3, line 38] As Adams et al. (2004) is a review of 10-years’ worth of work, it might be worth adding ‘and references therein’.

[Page 3, line 47] These two references are in neither alphabetical nor chronological order. If they refer to ‘convergent evolution’ and ‘adaptative radiation’ respectively, it is noticed they appear on the next page in the same order, although the order of convergence and adaptive radiation are reversed.

[Page 3, line 3] ‘…which convergently resemble’. The addition of ‘convergently’ makes this sentence is a little grammatically awkward as it /could/ be read that the tenrecs that resemble shrews, moles, and hedgehogs are convergent on each other rather than their lipotyphlan doppelgangers.

[Page 4, line 61] This sentence is a repetition.

[Page 4, line 67-72] ‘Morphological diversity… gain an insight into their evolution’. Feels repetitive from page 2.

[Page 4, 76] It is noted that although much is made of the assumed less diverse cranial anatomy of golden moles, no qualitative description of golden mole skull anatomy is presented. A brief overview of broad anatomical differences previously ‘qualitatively’ described from the literature (which, I assume was the germ of this paper) would be useful for a reader unassociated with tenrecid and chrysochlorid morphology. E.g. an illustrative diagram of a few species of tenrecs and golden moles indicating differences would perhaps help the reader - especially as there are no images of chrysochlorids in the paper regardless of them being referred to often.

[Page 5, 83] ‘though not always’. This is an important caveat and is an interesting avenue of research - but is not referenced. Perhaps this would be a good place to consider, for instance, the decoupling of functional convergence and anatomical convergence e.g. McGee and Wainwright (2012) doi:10.1111/j.1558-5646.2012.01839.x

[Page 5, line 99] ‘all of the… in the collections’. Irrespective of damage? This is referenced later in the methods, but would be useful to clarify here.

[Page 6, line 120] Please reference what image software was used.

[Page 7, line 126] ‘Available on request’. As this follows immediately after mentioning the museums, it is unclear whether the request should be made to the authors or the institutions.

[Page 7, line 131] Scale bar not referenced before this point as part of the photography protocol.

[Page 7, line 141] ‘depict that’ change to ‘depict the’.

[Page 7, 142- Page 8, 149] Consider moving this large block of references to the end of the sentence or after ‘where available’.

[Page 9, line 178] ‘that has relatively low morphological diversity’. Qualify that the Microgale genus has previously been qualitatively considered to have low diversity.

[Page 9, line 179-180] Reconsider this switch into present tense.

[Page 13, line 274] This is the fourth use of either ‘commonly cited’ or ‘often cited’.

[Page 14, line 300-301] ‘…particularly diverse palate morphologies’. Are these broad diets similar in terms of them being broad, or similar in terms of the actual breadth of foodstuffs that make up these ‘broad’ dietary preferences? Also, it may be worth mentioning teeth morphology here (obviously linked to diet) as the authors reference so many sources of anatomy between pages 7 and 8, and also any osteological correlates of jaw musculature as well as palate size/shape?

[Page 15, line 329-332] ‘the majority of tenrecs are very morphologically similar’. I suppose here the authors are talking about Microgale specifically, but if not, surely this sentence negates the thesis and, moreover, counters the repeated fact that tenrecs are ‘often cited as an example of an exceptionally morphologically diverse group?’ Especially as the authors reference a 2003 paper?

[Page 16, line 335-36] Do these three really need to be referenced here again? If so, they should be either chronologically or alphabetically ordered.

[Page 16, line 352] ‘…restricted to just one axis…’. Consider altering the word ‘axis’ here as it has connotations of the PCAs performed as part of this study rather than the scope of the morphological data collected - as I think is meant.

Figure 1
Some of the labels within the flow chart boxes might need tweaking. Some are descriptions of parts of the methodology (e.g. ‘Landmarks and curves’, ‘PC axes that account for 95% variation’) whereas others are instructions (e.g. ‘Combine data in R’, ‘Compare diversity of groups’).

Figure 2
It is a shame this diagram is a chimera of two specimens. Surely out of the vast data set there was one skull from which all three views could be taken?

‘1cm’ on scale bar has a space in the ventral view.

Due to the darkness of the photograph, the landmarks are difficult to see (esp. 8, 9, and 10). Consider a different colour or shade to show them more easily, or using a diagrammatic skull rather than a photograph.

The curves are very difficult to see. Consider changing colour or tone.

Figure 3
In text – ‘95%’ rather than ‘95\%’

Table 1
0.04 for 31 lateral is not bold.

Table 2
Although these are all significant, consider making all p values bold in keeping with Table 1.

Additional:
It is very obvious that some references are repeated many times. For instance, Soarimalala and Goodman (2011) is referenced no fewer than ten times throughout the paper. Although the authors are obviously very knowledgeable in all facets of tenrecid biology, this repetition sometimes results in a feeling that these references are cited instead of more salient works from before 2011 – a feeling amplified by the repeated twinning of Soarimalala and Goodman with a paper from 1969. A slightly more diverse bibliography would remedy this.

---

## Round 0.2 · accepted · Accept

Thank you for carefully attending to all the points raised by the reviewers and for providing a detailed response. I think this version is much improved and has fully addressed all those points. I am happy to accept the manuscript for publication.

I noted a few very minor points that should be corrected in the final version:

Page and line numbers from the pdf.
Pg5, ln97: would it be more accurate to write, “undamaged” rather than “intact”
Pg8, ln155: please insert a reference to the supplementary table at the end of this sentence
Pg8, ln164: I think it would be helpful if you could briefly state what this method is here rather than referring the reader only to the reference
Pg8, ln165: again, here, would be helpful to reference the supplementary table and/or S1
Pg12, ln246: use of “true”. Given the context, I would replace that with either “supported” or “not rejected”
Pg12, ln263-265: use of “sample size 12/31” I would use sample size of N=12 or something similar here

Fig 2A: placement of 2A appears to overlap with 2B such that part of the “8” is not visible in 2A